# Multiphysics Simulation of Crosstalk Effect in Resistive Random Access Memory with Different Metal Oxides

**DOI:** 10.3390/mi13020266

**Published:** 2022-02-06

**Authors:** Hao Xie, Jun Hu, Zhili Wang, Xiaohui Hu, Hong Liu, Wei Qi, Shuo Zhang

**Affiliations:** 1School of Information and Electrical Engineering, Zhejiang University City College, Hangzhou 310015, China; xieh@zju.edu.cn (H.X.); huxh@zucc.edu.cn (X.H.); liuhong@zucc.edu.cn (H.L.); 2College of Information Science and Electronic Engineering, Zhejiang University, Hangzhou 310027, China; 3Center for Optical and Electromagnetics Research (COER), Zhejiang University, Hangzhou 310058, China; hujun@zju.edu.cn; 4Science and Technology on Electromagnetic Compatibility Laboratory, China Ship Development and Design Centre, Wuhan 430064, China; wzl7616003@126.com

**Keywords:** finite difference method, graphene electrode, metal oxide, oxygen vacancy, resistive random access memory, thermal crosstalk

## Abstract

Based on the electrical conductivity model built for graphene oxide, the thermal crosstalk effects of resistive random access memory (RRAM) with graphene electrode and Pt electrode are simulated and compared. The thermal crosstalk effects of Pt-RRAM with different metal oxides of TiO_x_, NiO_x_, HfO_x_, and ZrO_x_ are further simulated and compared to guide its compatibility design. In the Pt-RRAM array, the distributions of oxygen vacancy density and temperature are obtained, and the minimum spacing between adjacent conduction filaments to avoid device operation failure is discussed. The abovementioned four metal oxides have different physical parameters such as diffusivity, electrical conductivity, and thermal conductivity, from which the characters of the RRAMs based on one of the oxides are analyzed. Numerical results reveal that thermal crosstalk effects are severe as the spacing between adjacent conduction filaments is small, even leading to the change of logic state and device failure.

## 1. Introduction

High performance nonvolatile memories have become a research hotspot because of the growing demands for data-intensive applications in areas such as artificial intelligence, big data, and internet of things [1,2,3,4,5]. Notably, resistive random access memory (RRAM) is one of those promising candidates for next-generation high-capacity data storage due to its cost-effective fabrication, high operating speed, long data retention, and low power consumption [6,7,8,9,10].

The mechanism of RRAM is the resistive switching through conduction filament growth (low-resistance state, LRS) and rupture (high-resistance state, HRS) in a resistance-switchable layer [11,12,13]. The commonly used materials for this unique layer are transition metal oxides, such as TiO_x_, NiO_x_, HfO_x_, ZrO_x_, TaO_x_, WO_x_, and AlO_x_ [14,15,16,17,18,19,20,21]. However, issues such as high thermal crosstalk effect, low integration density, and high power consumption in the metal oxides switching materials-based RRAMs remain to be solved [12,13,22].

In addition, the thermal crosstalk effect becomes a significant issue as the integration density of RRAM arrays increases. It may lead to the undesired change of logic state and device failure, which will directly affect the reliability of RRAM [12,13,21]. Some strategies such as improving the array structure, increasing the feature size, and exploiting two-dimensional materials (e.g., graphene and *h*-BN) were developed to overcome these problems [23,24,25,26,27,28]. Recently, graphene-electrode resistive random access memory (GE-RRAM) has been investigated experimentally [24], which can reduce the power consumption, variability of the set/reset voltage and current, and hence enhance the endurance of RRAM [24,29]. However, as far as we know, there are few studies on multiphysics performance comparison of the RRAM arrays with different electrodes or metal oxides. Therefore, a comprehensive computational study with fully coupled multiphysics is indeed necessary.

In this paper, the electrothermal characteristics of GE-RRAM, and Pt-RRAM are numerically studied and compared by fully coupled multiphysics simulations, which are performed by self-consistently solving the current transport, heat conduction, and oxygen vacancy migration equations. In addition, the Pt-RRAM with different metal oxides (TiO_x_, NiO_x_, HfO_x_, and ZrO_x_) are further simulated to guide its compatibility design. The paper is organized as follows. In Section 2, the device structure, modeling and simulation methods are discussed. Numerical results between GE-RRAM and Pt-RRAM, and Pt-RRAM with different metal oxides are presented and discussed in Section 3. Conclusions are finally drawn in Section 4.

## 2. Device Structure and Methodology

### 2.1. Device Structure

The geometry of GE-RRAM array is shown in Figure 1, where conduction filaments are formed between pillar metal and graphene electrodes. The detailed configuration and cross section of GE-RRAM cells are shown in Figure 1b,c, respectively. The insulating material is sandwiched between the neighboring graphene electrodes. During the reset process, a positive electric voltage is applied onto the pillar electrode, while a high or low electric voltage can be applied onto one graphene electrode to control the selected cell to be inactive or active [29].

Figure 2a shows the typical architecture of a Pt-RRAM array, in which conduction filaments are formed between the pillar and metal electrodes. The details and cross section of Pt-RRAM cells are shown in Figure 2b,c, respectively. An insulating material is filled between the adjacent metal electrodes. During the reset process, a positive electric voltage is applied onto the pillar electrode, and a high or low electric voltage is applied onto one metal electrode to control the selected cell to be inactive or active, respectively [13].

### 2.2. Multiphysics Progresses

A fully coupled multiphysics simulator based on finite difference method is developed to simulate the crosstalk effect in both GE-RRAM and Pt-RRAM arrays. The current transport, oxygen vacancy migration, and heat conduction processes are simulated by solving the coupled current continuity, oxygen vacancy drift-diffusion, and heat conduction equations [12,13,22,29]. The migration of oxygen vacancy mainly comprises the diffusion process caused by density gradient, and the drift process driven by electric field. It is highly temperature dependent, so its thermal effect should be carefully simulated and analyzed [30].

Oxygen vacancy migration equation is given by,
(1)$$\frac{\partial n\left(\vec{r},t\right)}{\partial t}=\nabla \cdot \left(D\left(\vec{r},t\right)\cdot \nabla n\left(\vec{r},t\right)-\vec{v}\left(\vec{r},t\right)\cdot n\left(\vec{r},t\right)\right)$$
where $n\left(\vec{r},t\right)$ is the oxygen vacancy density, $D\left(\vec{r},t\right)$ is the diffusion coefficient, and $\vec{v}\left(\vec{r},t\right)$ is the drift velocity.

The diffusion coefficient is,
(2)$$D\left(\vec{r},t\right)=D_{0}\cdot \exp(-E_{a}/k_{b}T\left(\vec{r},t\right))$$
where $E_{a}$, $k_{b}$, $T\left(\vec{r},t\right)$, and $D_{0}$ are the diffusion barrier, Boltzmann constant, temperature, and pre-exponential factor of diffusion, respectively.

The current continuity equation can be written as,
(3)$$\nabla \cdot \left(\sigma \left(\vec{r},t\right)\nabla V\left(\vec{r},t\right)\right)=0$$
where $\sigma \left(\vec{r},t\right)$ and $V\left(\vec{r},t\right)$ are the electrical conductivity and electric potential, respectively.

Besides, the heat conduction equation can be described as,
(4)$$\nabla \cdot \left(\kappa \left(\vec{r},t\right)\nabla T\left(\vec{r},t\right)\right)=-\sigma \left(\vec{r},t\right){\left|E\left(\vec{r},t\right)\right|}^{2}$$
where $\kappa \left(\vec{r},t\right)$ is the thermal conductivity, and $E\left(\vec{r},t\right)$ is the electric field.

The partial differential Equations (1), (3), and (4) are coupled through the parameters such as $n\left(\vec{r},t\right)$, $\sigma \left(\vec{r},t\right)$, $\kappa \left(\vec{r},t\right)$, and $T\left(\vec{r},t\right)$, can be solved self-consistently to simulate the multiphysics processes in the RRAM array.

### 2.3. Physical Parameters

Resistive materials are the core of RRAM, and different resistive materials have great impacts on the performance of RRAM. Due to its distinctive advantages including simple composition, simple preparation, and compatibility with CMOS process, metal-oxide-based RRAMs have been the research focus of famous semiconductor companies and research teams worldwide. The commonly used metal oxide materials include HfO_x_, TiO_x_, ZrO_x_, and NiO_x_. In order to study their power consumption and reliability properties, it is necessary to establish the multiphysics model of RRAMs with different metal oxide materials.

The thermal conductivity, electrical conductivity, and conduction activation energy $E_{AC}$ have dependence on $n\left(\vec{r},t\right)$ as shown in the Figure 3. The electrical conductivity of metal is described by Arrhenius equation as follows [22],
(5)$$\sigma \left(\vec{r},t\right)=\sigma_{0}\exp\left(-\frac{E_{AC}}{k_{b}T\left(\vec{r},t\right)}\right),$$
where $E_{AC}$ is active energy, and $\sigma_{0}$ is the coefficient of electrical conductivity, as given by,
(6)$$\sigma_{0}=\{\begin{array}{cc} \sigma_{s}\; & \;n=0\;\\ \frac{n}{n_{cf\_s}}*\sigma_{e}\; & \;0<n<n_{cf\_s}\\ \sigma_{e}\; & \;n\geq n_{cf\_s} \end{array},$$
where $n_{cf\_s}$ is the initial oxygen vacancy density of conduction filaments, $\sigma_{s}$ and $\sigma_{e}\;$ are the electrical conductivities at n=0 and $n\geq n_{cf\_s}$, respectively.

The thermal conductivity of metal oxide is given by,
(7)$$\kappa \left(\vec{r},t\right)=\kappa_{0}\cdot \left(1+\lambda \left(T-T_{0}\right)\right),$$
where λ is the temperature coefficient, $T_{0}=300\;K\cdot$$\kappa_{0}$ is the coefficient of thermal conductivity, which is described as follows,
(8)$$\kappa_{0}=\{\begin{array}{cc} \kappa_{s}\; & \;n=0\;\\ \frac{n}{n_{cf\_s}}*\kappa_{e}\; & \;0<n<n_{cf\_s}\\ \kappa_{e}\; & \;n\geq n_{cf\_s} \end{array}.$$
where $\kappa_{s}$ and $\kappa_{e}\;$ are the thermal conductivities at n=0 and $n\geq n_{cf\_s}$, respectively.

The key physical parameters of metal oxide materials HfO_x_, TiO_x_, ZrO_x_, and NiO_x_ are given in Table 1, which will be used for the multiphysics simulations of Ge-RRAM and Pt-RRAM, as discussed in Section 3.

Besides, the electrical conductivity of graphene oxide in GE-RRAM is highly dependent on its oxidation degree. An electrical conductivity model of graphene oxide built in [29] is given by Equation (9) and shown in Figure 4, and it is exponentially dependent on the density of oxygen atoms in the partially oxidized graphene oxide,
(9)$$\sigma_{ox}\left(\vec{r},t\right)=\sigma_{o}e^{\left(-r\times o\left(\vec{r},t\right)\right)}$$
where $\sigma_{o}$ is the electrical conductivity of pure graphene ($\sigma_{o}$ = 5 × 10^5^ Ω^−1^m^−1^), $o\left(\vec{r},t\right)$ is the oxygen atom density of graphene oxide, and *r* is a constant (r = 1.58 × 10^−18^ m^2^).

## 3. Simulation Results and Discussion

### 3.1. Multiphysics Simulation of GE-RRAM and Pt-RRAM

The oxygen vacancy drift-diffusion, current transport, heat conduction equations are solved self-consistently by the time-domain finite difference method, and the multiphysics effects between adjacent cells in GE-RRAM and Pt-RRAM arrays are further compared and analyzed. One bad case is considered in this paper, in which the victim inactive cell is sandwiched between two adjacent active cells. In both GE-RRAM and Pt-RRAM arrays, the applied voltage on the pillar electrode starts at 0 V and increases linearly at a rate of 1 V/s. The top and bottom planar electrodes (active cells) are grounded, but the voltage of middle planar electrode (inactive cell) stays the same as that of the pillar electrode. The temperatures of the top and bottom surfaces are set as 300 K, and all the other side surfaces are set with Neumann boundary conditions [29]. The adiabatic boundary condition is implemented on the rest of the boundaries in the simulation [29].

Figure 5 shows the initial oxygen vacancy density distributions of GE-RRAM and Pt-RRAM, in which both top and bottom cells are active while the middle cell is inactive. The spacing between the center of adjacent conduction filaments in GE-RRAM is 6 nm. The initial oxygen vacancy density of the conduction filament is 1.2 × 10^27^ m^−3^. The other parameters are set the same as in [29]. Here, the spacing is defined as the distance between the neighboring boundaries of adjacent conduction filaments. The diameter of the conduction filament in Pt-RRAM is 6 nm, and the thickness of the insulating layer is 6 nm. A depletion layer will be formed in the conduction filament to make resistance state switched from LRS to HRS, where the oxygen vacancy density of 0.6 × 10^27^ m^−3^ is defined as the threshold for the formation of the depletion layer [13,22]. The conduction filaments have been formed in the forming process, which exist at the beginning of the reset process. Even if the middle cell is inactive, its conduction filament still exists at the beginning of the reset process.

Figure 6 presents the oxygen vacancy density and temperature distributions in Pt-RRAM. The depletion layer has formed in the middle conduction filament of the inactive cell in Figure 6a, since the inactive cell was affected by the adjacent active cells. This may lead to the change of logic state and device failure. Due to the crosstalk effect from the adjacent active cells, the temperature of the inactive cell has reached 632 K. The temperature of active cells increases rapidly due to the Joule heating effect during the reset process. A depletion layer is gradually formed in the conduction filament of the active cells, leading to the increment of cell resistance. The initial high temperature region locates around the conduction filament of the active cells, since the larger current density. As the heat accumulates and the conduction filament breaks, parts of the heat transfer from above and below active cells to the middle inactive cell, which raise temperature in the victim cell and thermally activate the diffusion of oxygen vacancy to the adjacent low oxygen vacancy density region. Then, due to low local thermal conductivity of the low oxygen vacancy density region, the high temperature region locates at the victim cell. Further, the conduction filament of the victim cell gradually become thinner and lead to formation of depletion layer. Therefore, the thermal crosstalk effect of adjacent elements in the Pt-RRAM array is obvious.

In comparison, Figure 7 shows the oxygen vacancy density and temperature distributions in GE-RRAM. The corresponding applied voltage is 0.2 V. There is no obvious crosstalk effect between adjacent RRAM cells in this extreme case. During the reset process, oxygen ions are pushed back from the graphene electrode to metal oxide while a positive voltage is applied onto the pillar electrode, resulting in the rupture of conduction filament. An oxygen vacancy depletion layer is formed in the little part of oxide near graphene electrode because oxygen vacancies are occupied by oxygen ions from the graphene electrode. Meanwhile, temperature of the active cell is only 337 K, and the temperature of the middle inactive cell is barely affected by the above and below active cells. In the GE-RRAM, the resistive state variations of the conduction filaments are mainly realized by the rapid movement of oxygen ions in the metal oxides material and the graphene electrode. The spacing between adjacent cells can be set as small as possible to achieve a high degree of integration under the condition of current processing technology.

It can be seen that the thermal crosstalk effect between adjacent cells in Pt-RRAM array is much serve than that in GE-RRAM array. The above multiphysics simulations of Pt-RRAM and GE-RRAM arrays show that GE-RRAM can achieve higher integration, which can be used to guide their compatibility design.

### 3.2. Multiphysics Simulation of Pt-RRAM with Different Metal Oxides

In this paper, the thermal crosstalk effect of reset process are simulated, and its set process will follow the similar trend qualitatively [12]. In order to explore the multiphysics effects of Pt-RRAM and further guide its compatibility design, the reset voltage of active Pt-RRAM cell with different metal oxides is extracted and plotted in Figure 8. The reset voltage of HfO_x_-, TiO_x_-, ZrO_x_-, and NiO_x_-based Pt-RRAM is 0.2 V, 0.1 V, 0.18 V, and 0.14 V, respectively. It indicates that the reset voltages of Pt-RRAM with different metal oxides are different, since different diffusivity requires different electric fields to accelerate the diffusion transport of oxygen vacancies.

Figure 9 shows the oxygen vacancy density profiles along the center of middle inactive cell in Pt-RRAM with different metal oxides, while the applied voltages is their reset voltages. It is indicated that oxygen vacancy density is decreased in the conduction filament of the victim cell.

The crosstalk effect for different spacings between adjacent conduction filaments are further carried out, since the crosstalk effect depends more on the spacing than the width of conduction filaments [13]. Figure 10 shows that the minimum spacings between adjacent conduction filaments to avoid device failure in HfO_x_-, TiO_x_-, ZrO_x_-, and NiO_x_-based Pt-RRAM arrays is 7 nm, 14 nm, 9 nm, and 10 nm, respectively. With the increase of the spacing between adjacent conduction filaments, the degree of heat transfer from the above and below active cells to the middle inactive cell will decrease. The lower temperature of the victim cell, the smaller diffusion of its oxygen vacancy to the adjacent low oxygen vacancy density region, so as to further reduce the crosstalk effect of RRAM. Increasing the spacing between adjacent conduction filaments is a feasible way to mitigate thermal crosstalk effect. It indicates that the minimum spacings of TiO_x_ and HfO_x_ are the largest and smallest, respectively.

Figure 11 and Figure 12 show oxygen vacancy, and temperature distributions in TiO_x_- and HfO_x_-based Pt-RRAM arrays with the same spacing under different reset voltages, respectively. It indicates that the oxygen vacancy of the middle conduction filament is higher than 0.6 × 10^27^ m^−3^ in TiO_x_-based Pt-RRAM, which means the conduction filament stays intact. On the contrary, the depletion layer has formed in HfO_x_-based Pt-RRAM, since the oxygen vacancy of the middle conduction filament is lower than 0.6 × 10^27^ m^−3^. The highest temperatures of TiO_x_- and HfO_x_-based Pt-RRAM arrays are 800.9 K and 600.7 K in the victim cells, respectively. Due to the Joule heating effect during the reset process, the temperature of active cells increases rapidly. As the heat accumulates, parts of the heat transfer from active cells to the middle inactive cell, and raise the temperature in the victim cell. The rise of temperature thermally activates the diffusion of oxygen vacancy of the victim cell to the adjacent low oxygen vacancy density region, and finally leads to formation of the depletion layer in HfO_x_-based Pt-RRAM array.

## 4. Conclusions

In this paper, the reset processes of GE-RRAM and Pt-RRAM with four metal oxides are numerically studied and compared. Numerical results show the characteristics of different RRAMs: (1) The thermal crosstalk effect in GE-RRAM array is much smaller than that in Pt-RRAM array, and the GE-RRAM can achieve higher integration. (2) The reset voltages from big to small in Pt-RRAMs with different metal oxides are HfO_x_, ZrO_x_, NiO_x_, and TiO_x_. (3) The degree of thermal crosstalk effect from high to low in Pt-RRAMs with different metal oxides are TiO_x_, NiO_x_, ZrO_x_, and HfO_x_.

## Figures and Tables

**Figure 1 micromachines-13-00266-f001:**
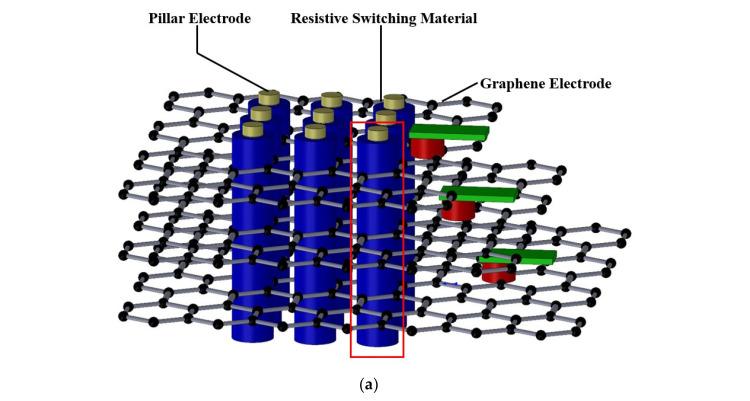
(**a**) Schematic of GE-RRAM array, (**b**) details of GE-RRAM cells in one pillar electrode, and (**c**) cross section of a single GE-RRAM cell through the central axis of the pillar electrode.

**Figure 2 micromachines-13-00266-f002:**
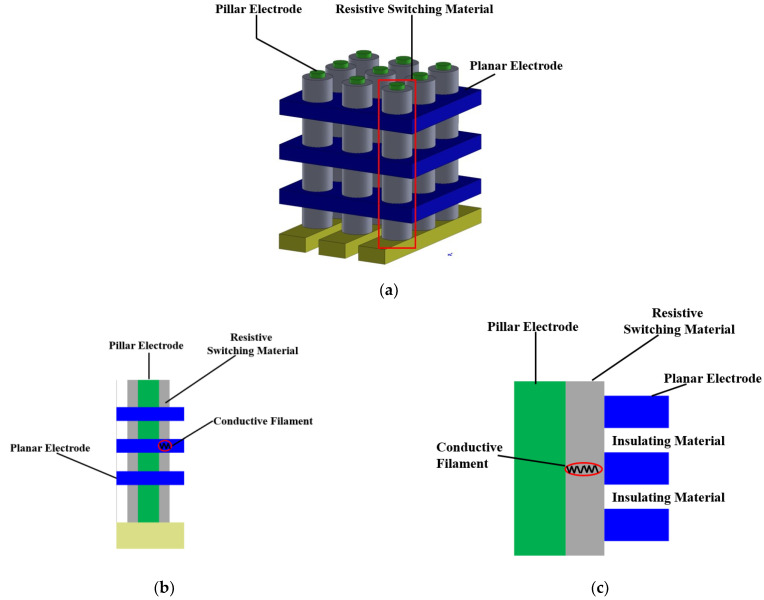
(**a**) Schematic of Pt-RRAM array, (**b**) details of Pt-RRAM cells in one pillar electrode, and (**c**) cross section of Pt-RRAM cells through the central axis of the pillar electrode.

**Figure 3 micromachines-13-00266-f003:**
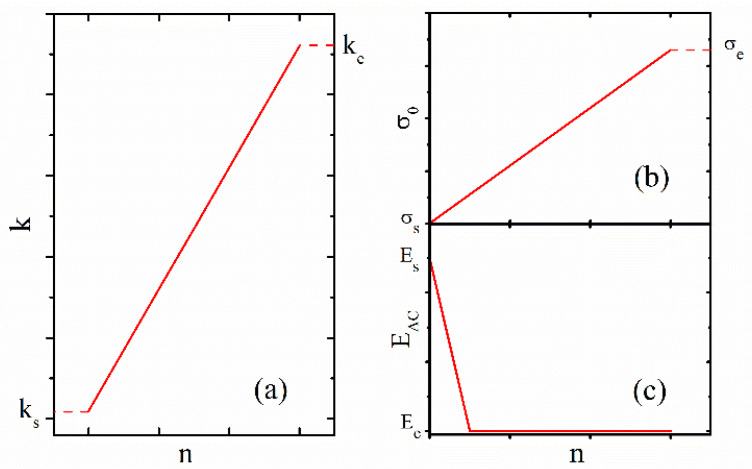
(**a**) Thermal conductivity, (**b**) electrical conductivity, and (**c**) conduction activation energy as a function of oxygen vacancy density [22].

**Figure 4 micromachines-13-00266-f004:**
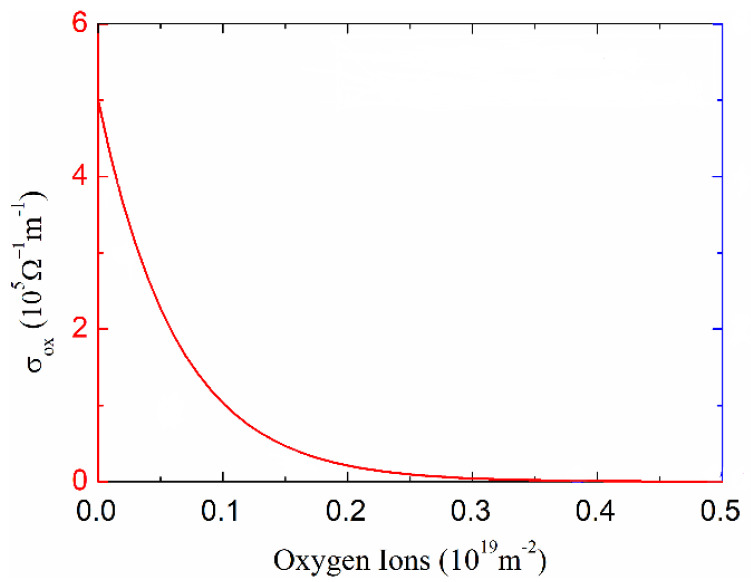
Electrical conductivity model of the partially oxidized graphene oxide [29].

**Figure 5 micromachines-13-00266-f005:**
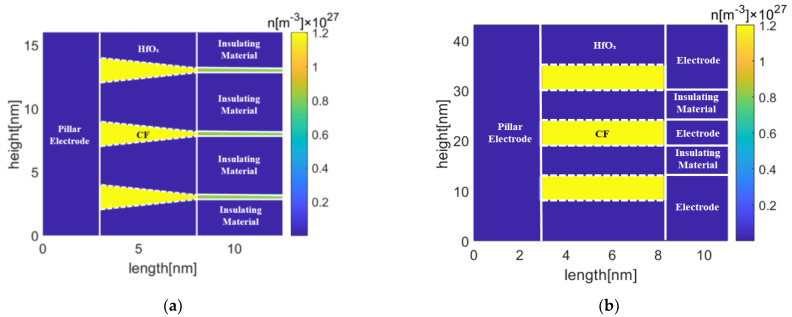
Oxygen vacancy density distributions for the (**a**) GE-RRAM and (**b**) Pt-RRAM, in which both the top and bottom cells are active, and captured at 0 s.

**Figure 6 micromachines-13-00266-f006:**
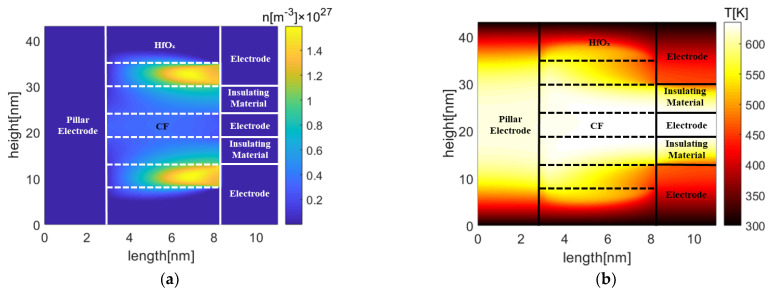
Distributions of (**a**) oxygen vacancy density and (**b**) temperature in Pt-RRAM. Both top and bottom cells are active, and captured at 0.2 s.

**Figure 7 micromachines-13-00266-f007:**
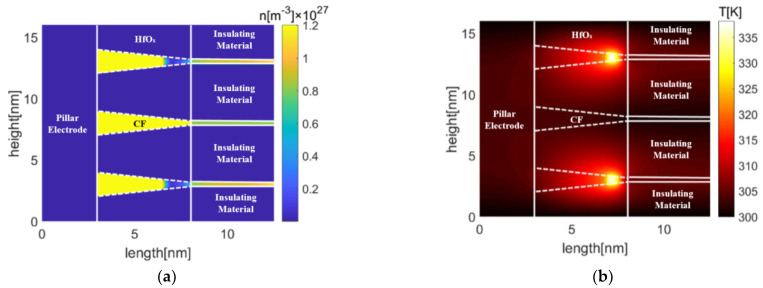
Distributions of (**a**) oxygen vacancy density and (**b**) temperature in GE-RRAM, where both the top and bottom cells are active, and captured at 0.2 s.

**Figure 8 micromachines-13-00266-f008:**
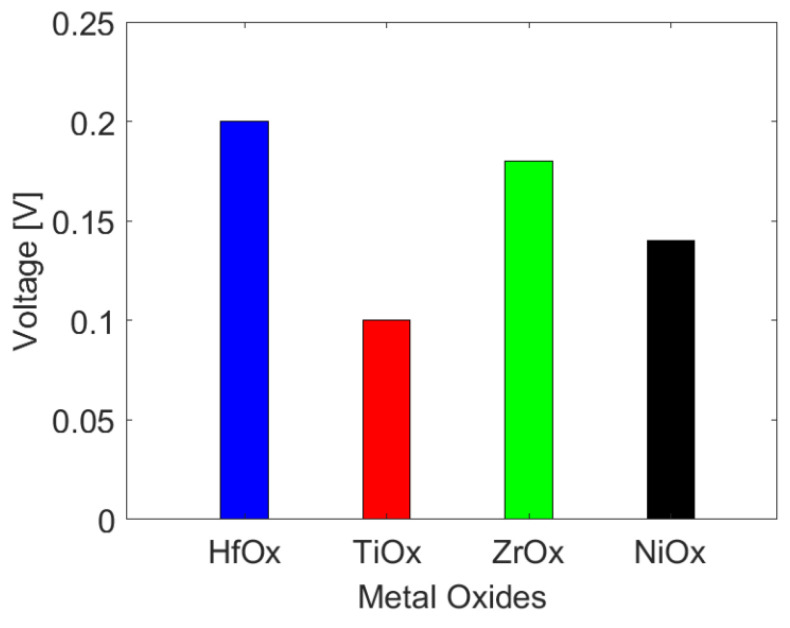
The reset voltage of Pt-RRAM with different metal oxides (HfO_x_, TiO_x_, ZrO_x_, and NiO_x_).

**Figure 9 micromachines-13-00266-f009:**
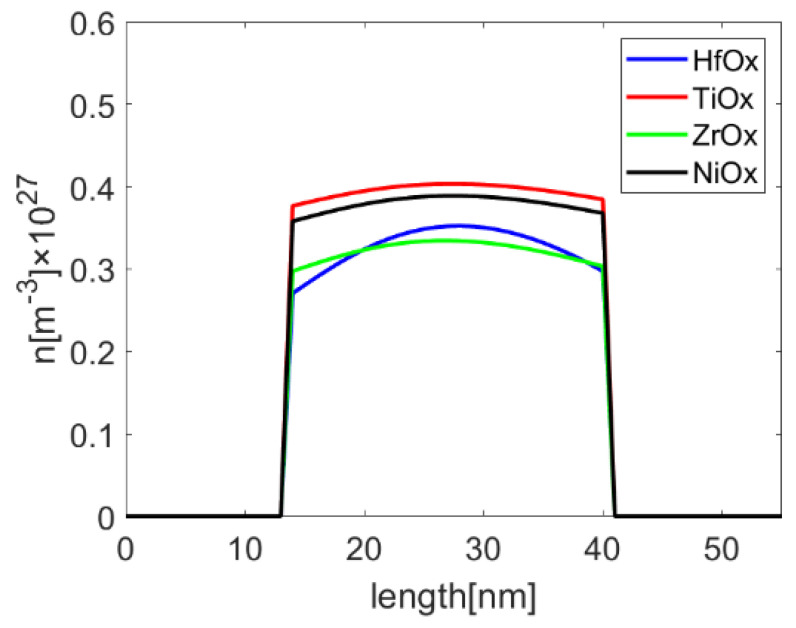
Oxygen vacancy density profiles along the center of inactive cell in Pt-RRAM. Both top and bottom cells are active, and captured under the condition of reset voltages.

**Figure 10 micromachines-13-00266-f010:**
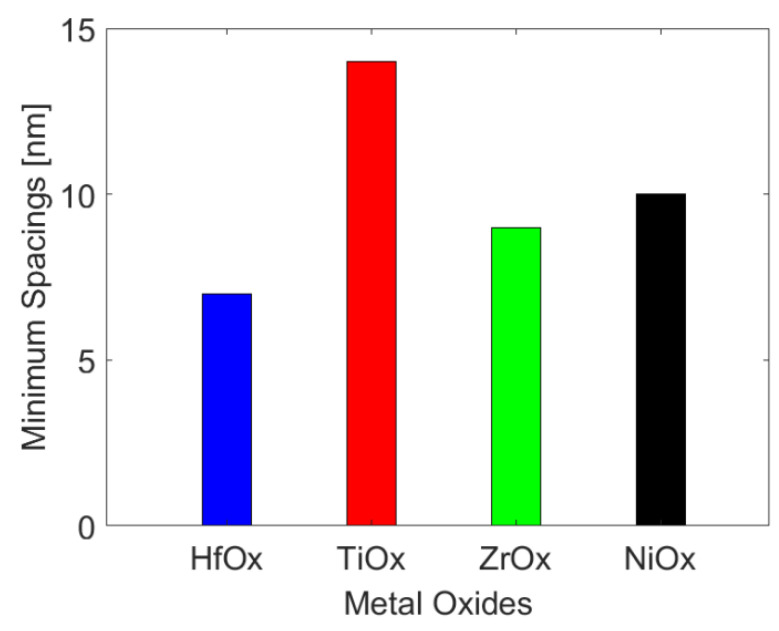
The minimum spacings between adjacent conduction filaments to avoid device failure in Pt-RRAM arrays with different metal oxides.

**Figure 11 micromachines-13-00266-f011:**
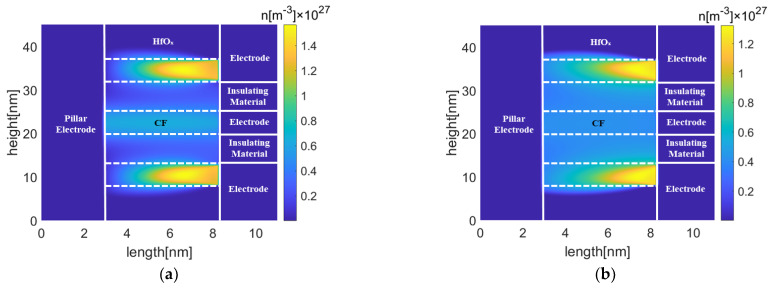
Oxygen vacancy distributions in Pt-RRAM arrays with the spacing of 7 nm (**a**) TiO_x_ (**b**) HfO_x_.

**Figure 12 micromachines-13-00266-f012:**
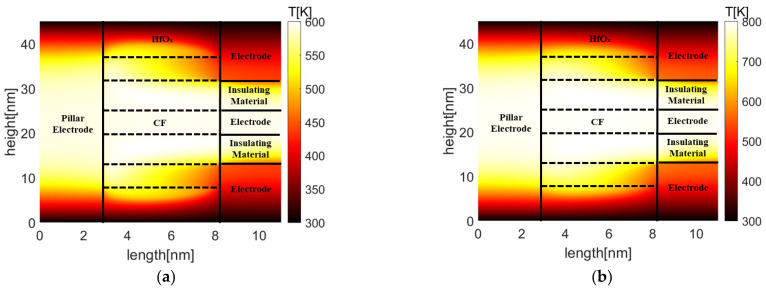
Temperature distributions in Pt-RRAM arrays with the spacing of 7 nm (**a**) TiO_x_ (**b**) HfO_x_.

**Table 1 micromachines-13-00266-t001:** Physical parameters of metal oxide materials [22,31,32,33,34,35].

Parameters/Materials	HfO_x_	TiO_x_	ZrO_x_	NiO_x_
$\kappa_{s}$ (W/(m∗K))	0.5	3	2	35
$\kappa_{e}\;\;$ (W/(m∗K))	23	22.5	22.5	91
$\sigma_{s}$ (S/cm)	10	0.5	0.1	0.1
$\sigma_{e}$ (S/cm)	3300	2.4 × 10^4^	8.5 × 10^3^	3.3 × 10^4^
$E_{AC\_s}$ (eV)	0.05	0.07	0.05	0.05
$E_{AC\_e}$ (eV)	0	0.02	0	0
$D_{0}$ (m^2^/s)	2 × 10^−7^	0.5 × 10^−7^	2.5 × 10^−7^	1 × 10^−7^
$E_{A}$ (eV)	1	1.1	1.5	1.5

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
