# Peer review of "Multiphysics Simulation of Crosstalk Effect in Resistive Random Access Memory with Different Metal Oxides"

_micromachines, 2022, doi:10.3390/mi13020266_

Round 1

Reviewer 1 Report

The manuscript “Multiphysics Simulation of Crosstalk Effect in Resistive Random Access Memory with Different Metal Oxides” has some issues that must be clarified. The comments are as follows”

  1. How can the thickness of the switching layer impact on the switching phenomena?
  2. The manuscript used the graphene as bottom electrode. Authors mentioned about the oxidizing of graphene electrodes. Is this effective for the practical application and how it can affect the device performance?
  3. In the Figure 5, the oxygen vacancy distribution is clear in the middle cell although the cell is inactive. Why there is a filament in the inactive middle cell? Is the cell first shifted to LRS before the top and bottom cell made active? Please explain it properly.
  4. Why the crosstalk effect of temperature (Figure 6b) in inactive cell is so high than the active cells in Pt-RRAM? What is the thickness of the insulation layer and how it can control the crosstalk effect?
  5. How can the RESET voltage affect the active GE-RRAM for different metal oxides? Is the thickness of the switching layer dependent on it?
  6. How the temperature affected the formation of depletion region in HfOx based Pt-RRAM? Also, what are the impact of highest temperature (~800°C) in TiOx devices on the depletion region?
  7. There are many mistakes present in the manuscript in terms of writing and choosing the English words. It must be rectified.

Reviewer 2 Report

This paper present the simulation works about GE-RRAM and Pt-RRAM with four metal oxides.  They study thermal crosstalk and reset voltage. I think this simulation work was well maded. can be extended into 3D VRRAM. Here some comments.

1) Why GE-RRAM is introduced ? need to add more motivation in introduction.

2) Why many difference between real devices and simulation voltage in Figure 8 ? Need to add references.

3) Why only consider reset voltage in this work? not set voltage?

4) Why highest Defect is located in the middle? need more explanation about thi.

Round 2

Reviewer 1 Report

Fine with revision